# Heterogeneity of Intestinal Tissue Eosinophils: Potential Considerations for Next-Generation Eosinophil-Targeting Strategies

**DOI:** 10.3390/cells10020426

**Published:** 2021-02-17

**Authors:** Joanne C. Masterson, Calies Menard-Katcher, Leigha D. Larsen, Glenn T. Furuta, Lisa A. Spencer

**Affiliations:** 1Gastrointestinal Eosinophilic Diseases Program, Department of Pediatrics, Digestive Health Institute, Children’s Hospital Colorado, University of Colorado School of Medicine, Aurora, CO 80045, USA; joanne.masterson@mu.ie (J.C.M.); calies.menard-katcher@childrenscolorado.org (C.M.-K.); leigha.larsen@cuanschutz.edu (L.D.L.); Glenn.Furuta@childrenscolorado.org (G.T.F.); 2GI and Liver Innate Immune Program, Department of Medicine, University of Colorado School of Medicine, Aurora, CO 80045, USA; 3Allergy, Inflammation & Remodeling Research Laboratory, Kathleen Lonsdale Institute for Human Health Research, Department of Biology, Maynooth University, Maynooth, County Kildare, Ireland

**Keywords:** eosinophil, eosinophilic gastrointestinal diseases, biologics, eosinophil sub-phenotypes

## Abstract

Eosinophils are implicated in the pathophysiology of a spectrum of eosinophil-associated diseases, including gastrointestinal eosinophilic diseases (EGIDs). Biologics that target the IL-5 pathway and are intended to ablate eosinophils have proved beneficial in severe eosinophilic asthma and may offer promise in treating some endotypes of EGIDs. However, destructive effector functions of eosinophils are only one side of the coin; eosinophils also play important roles in immune and tissue homeostasis. A growing body of data suggest tissue eosinophils represent a plastic and heterogeneous population of functional sub-phenotypes, shaped by environmental (systemic and local) pressures, which may differentially impact disease outcomes. This may be particularly relevant to the GI tract, wherein the highest density of eosinophils reside in the steady state, resident immune cells are exposed to an especially broad range of external and internal environmental pressures, and greater eosinophil longevity may uniquely enrich for co-expression of eosinophil sub-phenotypes. Here we review the growing evidence for functional sub-phenotypes of intestinal tissue eosinophils, with emphasis on the multifactorial pressures that shape and diversify eosinophil identity and potential targets to inform next-generation eosinophil-targeting strategies designed to restrain inflammatory eosinophil functions while sustaining homeostatic roles.

## 1. Introduction

Named by Paul Ehrlich in the late 1800s, eosinophils are primarily tissue-resident granulocytes whose physiologic functions have remained largely enigmatic. Associations of eosinophils and their cationic granule proteins with tissue damage, along with their presence in inflammatory exudates, have cast eosinophils as deleterious immune cells. However, pro-inflammatory effector functions of tissue eosinophils are increasingly being brought into balance with a growing recognition of their important contributions to immune, metabolic and tissue homeostasis, both locally and systemically, in health and disease. This apparent paradox is acutely seen within the gastrointestinal (GI) tract, where the highest density of eosinophils reside in the steady state, the cytokine milieu promotes exceptional longevity, and wherein eosinophils are implicated in immune and tissue homeostasis as well as in driving inflammatory eosinophilic gastrointestinal diseases (eosinophilic diseases (EGIDs); eosinophilic esophagitis (EoE), gastritis (EG), enteritis (EE) and colitis (EC)). Eosinophils are associated with multiple GI diseases in addition to the EGIDs, including gastric and colorectal cancers, and inflammatory bowel diseases (IBDs; Crohn’s disease and ulcerative colitis). In each of these disease settings parsing functional contributions of intestinal eosinophils is complex, as their presence may be associated with ameliorating and/or exacerbating tissue inflammation.

As already established for many other leukocytes, increasing evidence in health and disease suggest multifactorial pressures shape and diversify tissue eosinophils into distinct sub-phenotypes, oftentimes co-expressed within the same tissue. Should distinct functional subgroups of tissue eosinophils fulfill opposing inflammatory and homeostatic roles, then biologics designed to globally deplete eosinophils may be counter-indicated. The intent of this review is to synthesize the growing evidence for functional sub-phenotypes of intestinal tissue eosinophils, with emphasis on potential targets to inform next-generation eosinophil-targeting strategies designed to restrain inflammatory eosinophil functions while sustaining homeostatic roles. It has yet to be determined whether eosinophil sub-phenotypes represent transcriptionally-defined subsets, akin to T helper cell or macrophage subsets. Likewise, the degree to which eosinophils might revert between sub-phenotypes has yet to be established. For the purpose of this review, we will use the terms “subgroups” or “sub-phenotypes” interchangeably to broadly represent eosinophils that exhibit phenotypic distinctions.

## 2. Diverse Functions of Eosinophils in Intestinal Health and Disease

### 2.1. Terminal Effector Functions

Earliest observations of eosinophils dating back to the 1800s were in the context of parasitic helminth infections, tissues and inflammatory exudates from allergic diseases, and reactions to medications (reviewed in [1]). Despite recognitions of inflammation-dampening functions of eosinophils [2], obvious associations of eosinophils with tissue damage had supported a dominating historical view of eosinophils as primarily terminal effector cells. Pro-inflammatory effector functions of eosinophils derive from their storage and active degranulation of strongly basic proteins such as major basic protein (MBP), eosinophilic cationic protein (ECP), and eosinophil peroxidase (EPX) that directly damage parasite cuticles or inflamed tissues. Eosinophils may further exacerbate inflammation through secretion of pro-inflammatory cytokines; extrusion of extracellular DNA traps; deposition of Charcot–Leyden crystals; and release of reactive oxygen species through respiratory burst [3,4,5,6,7,8,9,10,11].

### 2.2. Steady-State Homeostasis

Eosinophils also play an underappreciated and understudied role in both tissue homeostasis and anti-inflammatory processes. Eosinophils accumulate in inflammatory states and thus are often considered “guilty by association,” however eosinophils are also present in select tissues during homeostasis, including the thymus, mammary gland, uterus, adipose tissue and intestine, with the highest levels in the GI tract during homeostasis [12]. Eosinophil homing to non-inflamed tissues at baseline suggests biological roles for eosinophils extend beyond end-stage effector functions. Homeostatic tissue eosinophils are implicated in tissue homeostasis and regeneration, epithelial barrier integrity, immune surveillance, and systemic regulation of glucose metabolism (reviewed in [4,13]). These observations prompted Lee and colleagues to propose an alternate conceptual framework, termed the LIAR (Local Immunity And/or Remodeling/Repair) hypothesis, that posits eosinophils are primarily tissue repair cells, attracted to areas of focal surges of cell death and proliferation in an attempt to restore tissue homeostasis [14].

In the GI tract eosinophils take up residence in the small and large intestines prenatally, suggesting a potential role in intestinal health surveillance and innate immunity [15]. To this end, eosinophils express pattern recognition receptors (PRRs), secrete anti-microbial and anti-viral factors, and trap and kill bacteria in extruded DNA nets, contributing to innate immune protection [7,16,17,18]. In mice, absence of intestinal eosinophils is associated with impaired Peyer’s patch development and reduced mucus layer [19,20,21,22], with depletion of small intestinal IgA observed in several [19,20,22], but not all [21], studies. Eosinophils have also been shown to play important roles in regulating the adaptive immune response. Bone marrow and intestinal eosinophils secrete critical survival factors necessary for support of long-lived plasma cells in the bone marrow and intestine [19,23,24]. Eosinophil-derived mediators contribute to T cell recruitment and modulation, and regulate intestinal dendritic cell function, indirectly influencing T cell activation [25,26,27,28,29]. Intestinal eosinophils also appear to be uniquely equipped to restrain intestinal inflammation. IL-1β is a pro-inflammatory cytokine implicated as a key mediator of inflammatory diseases including IBDs, in part due to its role in Th17 cell differentiation and maintenance [30,31]. Unlike blood or bone marrow eosinophils, eosinophils isolated from the small intestine of mice spontaneously secrete IL-1 receptor antagonist (IL-1Ra), a natural inhibitor of IL-1β. Frequencies of small intestinal eosinophils are inversely correlated with frequencies of intestinal Th17 cells, thus intestinal eosinophils play a physiological role in raising the threshold to induce Th17-associated inflammation in the steady state intestine [26].

### 2.3. Anti-Inflammatory Functions of Eosinophils within the Inflamed Gut

Excessive infiltrations of activated and degranulating eosinophils are associated with tissue damage in intestinal inflammatory diseases such as EGIDs and IBDs [13,32]. However, several lines of evidence suggest intestinal eosinophils are also important players in resolving active intestinal inflammation. In the context of mouse models of acute self-resolving inflammation, eosinophils have been elucidated as key players in lipid-mediated pro-resolving processes [33]. In particular, the pro-resolving lipid mediator Protectin D1 (PD1) is derived from the polyunsaturated fatty acid (PUFA) docosahexaenoic acid (DHA, 22:6 n-3) via the critical production of 12-/15-lipoxygenase enzymes by eosinophils. This has been reported in both mucosal (colonic, pulmonary) and non-mucosal (peritoneal) models of inflammation [34,35,36]. Despite this anti-inflammatory contribution in acute settings, mice experiencing more chronic models of small intestinal inflammation treated with anti-eosinophil (anti-IL-5 and anti-CCR3) antibodies experienced an attenuation of inflammation [37]. These almost discordant appearing studies likely indicate the differential contribution of eosinophils in acute versus more chronic inflammatory milieu. Interestingly, Miyata et al. discovered that while eosinophils in the periphery of control healthy subjects were capable of making 12-/15-lipoxygenase, patients with chronic poorly controlled asthma were deficient in their capacity to assist in the synthesis of Protectin D1 [35]. Overall, this growing body of research supports the potential for eosinophils to act in anti-inflammatory ways and one could speculate whether all eosinophils are made equally or indeed respond equally to systemic or tissue-specific micro-environmental signals. Further studies are indeed warranted to elucidate if this may be the case.

## 3. Eosinophil Heterogeneity

Although our central purpose here is to review the evidence for eosinophil heterogeneity specifically in the GI tract, we include discussions of blood and lung eosinophils as well, as the former represent the phenotype of eosinophils upon entry into intestinal tissues and the latter organ provides the most comprehensive understanding to date of the co-existence of eosinophil functional subtypes.

### 3.1. Blood Eosinophils

Eosinophils are primarily tissue-dwelling cells. The circulating pool of eosinophils represents a short-lived transitory population as they emerge from the bone marrow and traffic to tissues. Due to the relative ease of isolating eosinophils from peripheral blood compared to isolations from tissues, blood eosinophils have served as the source for a majority of eosinophil mechanistic studies in humans, and as a primary clinical outcome in evaluating efficacy of eosinophil-targeting biologics. Phenotypic analyses of blood eosinophils have focused predominantly on surface receptor expression and preformed cytokine content.

Blood eosinophils are broadly identified by flow cytometry as SSC^hi^SiglecF/8^+^CD62L^+^CCR3^+^IL5Rα^+^ leukocytes (reviewed in [4]), and express surface integrins including α_L_β_2_ (CD11a/CD18), α_M_β_2_(CD11b/CD18d), α_D_β_2_ (CD11d/CD18), α6 (CD49f), α_4_β_7_ (CD49d/β_7_), and β_1_ (CD29) integrins (e.g., α_4_β_1_) in low-activation conformations in the steady state [38,39]. Whether systemic signals within the context of eosinophilic inflammatory diseases may temporarily modulate surface receptors on circulating eosinophils, including integrin expression, activation markers and growth factor receptors, is an area of active current research, with most progress to date coming from studies in asthmatics. For example, within 48 h of bronchial segmental allergen challenge in asthmatic subjects, circulating eosinophils adopt a phenotype of enhanced α_D_β_2_ expression, and conversion of β_1_ integrins into an active conformation. These “activated” eosinophils within circulation are more prone to arrest on endothelial-expressed VCAM-1 [39], and correlate with decreased lung function in subjects with mild asthma [40]. A similar scenario of activation of eosinophil-expressed β1 integrins appears to occur in EoE patients [41] and may correlate with disease activity [42]. Moreover, reports of two case studies demonstrated a higher percentage of CD274^+^ (PDL1) blood eosinophils in EoE with reduction back down to normal levels following successful treatment [43]. Further studies are needed to determine the extent to which modulations of eosinophil surface marker profiles might be associated with EGIDs.

A dense population of intracellular granules containing an array of cationic proteins [3] and numerous cytokines [44,45,46] is among the most distinguishing features of eosinophils. In addition to de novo-synthesized cytokines, many functions of eosinophils in health and disease derive from their capacity for rapid and differential secretion from the diverse repertoire of pre-formed granule-derived cytokines [4,47,48], therefore the preformed cytokine profile of circulating eosinophils may impact their functional potential upon entry into target tissues. Bulk analysis of the granule cytokine contents within blood eosinophils from healthy donors revealed that eosinophils emerge from the bone marrow already stocked with a highly diverse cache of cytokines associated with Th1-, Th2- and regulatory functions, which are rapidly and differentially secreted in response to alterations in the cytokine milieu [49]. Relative concentrations of several of these preformed, granule-stored cytokines appear to be remarkably well conserved across different subjects (e.g., IL-5, IL-4 and IL-13), while other preformed cytokines exhibited moderate (e.g., IFN-γ, IL-8 and IL-10) or profound (e.g., IL-16) variability between individuals [50,51]. Applying cluster analysis, Ma etal. demonstrated at least some eosinophil-associated cytokines (e.g., IL-1β, IL-1α and IL-6) appear to be coordinately regulated [51]. How the preformed cytokine repertoire is shaped in human blood eosinophils remains elusive; however, eosinophil IL-16 content in a subset of normal donors correlated with body mass index [51], suggesting systemic metabolic factors may contribute to eosinophil cytokine profiles. Further studies are needed to determine whether and how environmental pressures modulate the surface receptor profile and shape the preformed cytokine repertoire of eosinophils within circulation in health and disease, how accurately mouse eosinophil granule contents recapitulate the preformed repertoire of human eosinophils, and whether these transient modulations might serve as biomarkers of disease status and/or identify functionally distinct subgroups.

### 3.2. Lessons from the Murine Lung

To date, the most comprehensive examples of the co-existence of eosinophil functional subtypes within tissues come from studies in the murine lung. Eosinophils sparsely populate the lung parenchyma of humans and mice in the steady state [52,53]. In mice, resident lung eosinophils exhibit a ring-shaped nucleus and a Siglec-F^int^CD125^int^CD62L^+^CD101^lo^CD11c^−^ surface phenotype resembling circulating blood eosinophils, express a transcriptome that includes several genes implicated in immune regulation and tissue homeostasis, and inhibit the maturation of allergen-loaded dendritic cells, thereby actively dampening Th2 immunity [53,54,55]. Within the context of allergic airway inflammation in humans and mice, homeostatic resident lung eosinophils are joined, and eventually outnumbered by, a distinct eosinophil subtype exhibiting an inflammatory phenotype [53]. In mice, “inflammatory” lung eosinophils are found in cell clusters within parenchyma, perivascular and peribronchial regions, exhibit increasingly segmented nuclei, adopt a Siglec-F^hi^CD125^int^CD62L^−^CD101^hi^CD11c^+^ surface phenotype and pro-inflammatory transcriptome, and fail to restrain Th2 immune responses [52,53,54]. Adding further support to the co-existence of functionally distinct eosinophil subtypes within the lung, Percopo etal. identified a small Gr1^hi^-expressing subset of lung eosinophils in allergen challenged or viral infected mice that express a distinct granule-derived cytokine repertoire, including expression of the B cell-active cytokines CXCL13 and IL-27, and high levels of IL-13 [56]. Of note, it remains unclear if the so-called “inflammatory” eosinophils uniformly promote inflammatory outcomes, as immunomodulatory functions have also been attributed to allergen-recruited eosinophils in murine models allergic airway diseases [28,57].

Taken together, lessons from the lung reveal baseline tissue-resident eosinophils, and eosinophils recruited within the context of allergic provocation, are functionally and phenotypically distinct and play dichotomous roles in maintaining homeostasis and disease exacerbation, respectively. Drawing from these and other observations, Abdala-Valencia and colleagues proposed a classification system for eosinophil sub-phenotypes (including distinguishing morphologic and surface receptor profiles) categorized by the morphogenetic state of the tissue. Their framework classifies tissue eosinophils as: (1) EoP (eosinophil progenitors); (2) Steady-state (homeostatic eosinophils resident within quiescent tissue at baseline); (3) Type 1 (interstitial eosinophils in transiently morphogenic tissues or type 1 immunity); or (4) Type 2 (epithelial-associated eosinophils recruited within the context of type 2 immunity) [58]. Studies are needed to determine the extent to which this evolving classification system might describe eosinophils resident within intestinal tissues in health and disease.

## 4. Heterogeneity of Intestinal Tissue Eosinophils

The intestinal tract provides a unique tissue setting within which to evaluate eosinophil sub-phenotypes in situ and to parse their potential differential impacts in inflammatory diseases. First, eosinophils home naturally to all regions of the GI tract at baseline, with the exception of the esophagus, and intestinal eosinophils are implicated in both homeostasis of the GI tract and disease exacerbation. Second, the cytokine milieu of the intestinal tissue environment uniquely promotes eosinophil longevity, which may support a fuller co-expression of phenotypic subpopulations.

### 4.1. Phenotypic Heterogeneity of Intestinal Tissue Eosinophils

#### 4.1.1. Phenotypic Heterogeneity of Intestinal Eosinophils in the Steady State

Eosinophils colonize the gastrointestinal tract prenatally independent of commensal bacteria and reside in both the small and large intestines throughout adulthood, with eosinophil densities decreasing from stomach to rectum [15,55,59]. Emerging data are revealing differences in eosinophil morphology, surface phenotype and localization within the GI tract that may collectively underly functionally distinct and differentially targetable eosinophil sub-phenotypes. Eosinophil phenotypic diversity is observed in comparing compartmentalized eosinophils at the macro level (e.g., blood versus intestinal tissue; small intestine versus colon) and also within more localized tissue microdomains (e.g., surrounding crypts versus villus-associated). The majority of available data derive from studies in mice. Therefore, discussions in this section will be focused on the mouse, unless otherwise indicated.

Surface receptor profiles of intestinal eosinophils distinguish them from bone marrow, blood and baseline lung eosinophils. Like uterine and thymic eosinophils (but notably distinct from baseline homeostatic lung eosinophils), the majority of intestinal eosinophils constitutively express CD11c [55,60], an adhesion molecule often used as a marker of dendritic cells, but also expressed on tissue macrophages and some subsets of Th2 cells. Ligands for CD11c include the intercellular adhesion molecule (ICAM) family of receptors, provisional extracellular matrix components (e.g., fibrinogen, laminin, collagen), complement fragments (iC3b), and bacterial cell walls (e.g., LPS) (reviewed in [61]). Compared to blood eosinophils, intestinal eosinophils also exhibit higher levels of Siglec F (Siglec 8 in humans) and CD11b [26,55,60]. The integrin CD11b interacts with a wide range of ligands, including periostin [62], an extracellular matrix protein strongly implicated in allergic diseases, including EGIDs [63]. Notably, the SiglecF^int/hi^CD11b^+^CD11c^+^ phenotype of intestinal eosinophils at baseline resembles recruited inflammatory eosinophils within lung tissue and bronchoalveolar lavage fluid of asthmatic humans and mice [53,54], suggesting that the local intestinal milieu maintains eosinophils in a relatively activated state, even at baseline. Intestinal eosinophils also exhibit higher levels of the common γ chain (γ^c^, utilized by cytokine receptors including. IL-2, IL-4, IL-7, IL-9, IL-15 and IL-21), which contribute to their selective survival [55], and GM-CSF receptor α (Rα). Additional receptors that distinguish intestinal from circulating eosinophils include inhibitory receptors (e.g., CD22 and SIRPα) that may serve to balance constitutive activation and survival signals (see Section 4.2.3 below). Eosinophils from the small intestine versus large intestine further differ in their expression of surface receptors, including CD11c, ST2, Ly6C and CD22 [19,64].

Within the small intestine, varied morphologies and surface receptor expression profiles suggest heterogeneity between eosinophils localized to the lamina propria (LP) surrounding crypts, compared to LP eosinophils within villi (Figure 1). Morphologically, small intestine-resident eosinophils in rats and mice within the LP surrounding crypts exhibit annular nuclei and rounded cell bodies with few, if any, short dendritic extensions [52,60,65]. Eosinophils in comparable submucosal locations within the cecum and colon appear more spread and exhibit numerous motile projections [52]. The minority subset of intestinal eosinophils that are CD11c^−^ localize to LP surrounding crypts, suggesting this region may represent a common entry point for blood eosinophils [65]. Building on the LIAR hypothesis it is feasible that the low-level morphogenetic activation associated with epithelial turnover at the crypt base may contribute to steady-state eosinophil homing to regions surrounding crypts (Figure 2). In contrast, eosinophils within LP of small intestinal villi exhibit spheric nuclei, broader cell bodies, more commonly exhibit cellular protrusions, and are uniformly CD11c^+^ [60,65]. Eosinophils with elongated cell bodies are frequently observed lying along the basement membrane of villus epithelium, extending dendritic processes into intraepithelial spaces [60,65]. Although rarely captured in histology sections, flow cytometry of tissue digests has confirmed eosinophils are present within the intraepithelial (IE) compartment at baseline in mice and humans [59,60]. Compared to LP eosinophils, IE eosinophils exhibit higher levels of CD11b, Siglec F and CD11c [60]. The emerging picture of a graded activation phenotype of small intestinal eosinophils, moving from regions surrounding crypts to villus epithelium, is reminiscent of the movement of airway eosinophils toward and across respiratory epithelia, and may have important clinical implications. Collectively, these data suggest that binary “tissue-resident homeostatic” versus “recruited inflammatory” designations evolving from studies in murine lungs are not likely to fully capture the complexities of co-resident intestinal eosinophil sub-phenotypes.

#### 4.1.2. Phenotypic Heterogeneity in Disease-Associated Microenvironments

Akin to eosinophils recruited to asthmatic airways, Vimilathas et al. demonstrated increased expression of total high-activity conformation αM (CD11b) integrin on esophageal eosinophils from EoE patients, expression of which promoted eosinophil survival through interactions with periostin [38]. Ultrastructural studies demonstrate esophageal tissue eosinophils undergo distinct modes of secretion (i.e., piecemeal degranulation versus cytolysis), the latter presumably indicative of pro-inflammatory functions, in patients with EoE [66]. In addition to phenotypic diversity, spatial redistributions of intestinal eosinophils (even in the absence of increased recruitment) also accompany disease settings; for example, allergen ingestion elicits redistribution of small intestinal eosinophils into villus LP [67], and in IBD (but not control) patients, actively degranulating eosinophils associate with cholinergic fibers [68]. Eosinophils also display significant metabolic flexibility [69,70], and further studies are needed to investigate contributions of metabolic plasticity to eosinophil functional sub-phenotypes.

Transcriptional profiles of intestinal eosinophils in steady state versus disease suggest eosinophil plasticity is associated with functional heterogeneity, shaped by the immediate tissue microenvironment. Reichman et al. compared transcriptional profiles of primary intestinal eosinophils at baseline, during active inflammation, and during resolution/repair phases in a murine model of dextran sulfate sodium (DSS)-induced colitis [71]. Eosinophils from the inflamed colon exhibited transcriptome signatures highlighted by upregulation of pro-inflammatory cytokines and chemokines, hallmark alarmins, inhibitory receptors (e.g., PIRB, SIRP1a), and transcripts associated with NADPH oxidase activity. In contrast, eosinophils from the colons of mice during resolution and repair phases had substantially downregulated their expression of proinflammatory cytokines [71]. In mouse models of acute or chronic bacterial infections, recruited eosinophils activated an IFN-γ-dependent transcription program, including upregulation of surface-expressed PD-L1. PD-L1-expressing eosinophils educated at the site of bacterial infection restricted immunopathology by dampening Th1 immune responses and associated tissue inflammation. These studies have led to the suggestion that PD-L1 expression might serve as a useful marker for colonic eosinophils with immune regulatory properties [72].

Distinctive transcriptional and proteomic profiles are also observed in eosinophils within tissue microdomains. Histologic surveys suggest tumor-associated eosinophilia tends to be associated with improved prognoses in patients with solid tumors, particularly in gastric and colorectal cancers [73]. Eosinophils are actively recruited to developing tumors and promote anti-tumor immunity in inflammation-induced and genetic spontaneous mouse models of colorectal cancer [73,74]. The tumor microenvironment supports prolonged eosinophil survival [74], and intratumor eosinophils exhibit pro-inflammatory gene and proteome signatures distinct from those of colonic eosinophils from tumor-adjacent tissue. The intratumor eosinophil transcriptome is characterized by genes involved in interferon signaling, enzymatic regulators of the extracellular matrix, and chemokines and cytokines implicated in recruitment of activated T cells [73,74]. The altered transcriptional profile of intratumor eosinophils was driven by granulocyte macrophage-colony stimulating factor (GM-CSF), and exogenous GM-CSF further enhanced the anti-tumor immune response [73]. Although multiple studies agree that intratumor eosinophils promote anti-tumor immunity, there are conflicting reports on the precise eosinophil-dependent mechanism(s), which may involve direct cytotoxic activities of eosinophils on tumor cells [74] and/or enhancing the Th1 anti-tumor immune response [73]. The latter finding is particularly intriguing in light of the tendency of intestinal eosinophils to dampen Th17 and Th1 in the steady state [26], and within the context of bacterial infection [72], respectively, and further supports the broad immune plasticity of intestinal tissue eosinophils, shaped by the local microenvironment.

Modulation of intestinal eosinophil surface receptor expression, spatial organization within the tissue, secretion status, metabolic profile and/or transcriptome in the context of disease may provide important insights into eosinophil functional phenotypes and may enable more meaningful evaluations of patient biopsies and samples that add qualitative information beyond “peak eosinophil counts.”

### 4.2. Local and Systemic Influences Shaping Intestinal Eosinophil Phenotype and Function

#### 4.2.1. Recruitment, Survival and the Intestinal Tissue Milieu

Steady-state homing of eosinophils into the small intestine is largely regulated by constitutive IL-5 secretion from resident ILC2 cells and eotaxin-1 secreted from epithelial cells and macrophages, with eotaxin acting through eosinophil-expressed CCR3 [15,75,76,77]. Although expendable for small intestine homing, eosinophil-expressed β7 integrin further contributes to eosinophil homing to the large bowel [78]. In type 2 inflammation, enhanced ILC2-derived IL-13 further drives macrophage and epithelial-derived secretion of eotaxins to exacerbate eosinophil recruitment and accumulation [77] (Figure 2). Normally devoid of eosinophils, IL-5 and eotaxins 1 and 3 also promote eosinophil infiltration into the esophagus in EoE [79]. Eotaxin stimulation dose-dependently elicits piecemeal degranulation in human blood eosinophils in vitro [80,81,82], suggesting eotaxin-mediated recruitment of intestinal eosinophils may contribute to the constitutive activation status of intestinal tissue eosinophils. Coordinated contributions of additional chemokines in shaping the spatial distribution within intestinal LP remain to be fully delineated.

Within intestinal LP, the cytokine microenvironment further shapes the activation phenotype of eosinophils in steady state and disease. Much of the putatively activated surface receptor phenotype of intestinal eosinophils can be recapitulated by exposure to IL-5 or GM-CSF. IL-5 and GM-CSF, along with IL-3, form a triad of important eosinophil priming and activation factors whose different heterodimer receptors signal through a shared common beta (β_c_) chain [83]. Signaling through β_c_ primes eosinophils, lowering their activation threshold for distinct modes of degranulation and oxidative functions; promotes longevity; and modulates surface Fc receptor and integrin expression profiles, enhancing engagement with extracellular matrix (ECM) components. Eosinophil responsiveness to these IL-5 family growth factors is limited in part by regulating expression of the respective cytokine-binding receptor α (Rα) chains. IL-5 signaling downregulates IL-5Rα in a negative feedback loop. In contrast, IL-5, GM-CSF and IL-3 upregulate eosinophil expression of GM-CSFRα and IL-3Rα, enabling comparatively longer-lived responsiveness to IL-3 and GM-CSF [84,85]. GM-CSF, secreted by epithelial cells and Paneth cells in addition to immune cells, is abundantly expressed in the small intestine (less so in the healthy colon) [86], promotes eosinophil adhesion and migration on ECM components at a lower active concentration than IL-5 or IL-3 [87], and likely plays a significant role in shaping the activation phenotype of small intestinal tissue eosinophils in the steady state [26]. Moreover, GM-CSF over-expressed within the colo-rectal tumor microenvironment drove alterations in the transcriptome of lesion-derived eosinophils [73] (see Section 4.1.2 above). Importantly, eosinophils themselves secrete growth factors and cytokines, including IL-5 and GM-CSF, suggesting intestinal eosinophils may promote their own survival and activation through autocrine signaling. Soluble (but not immobilized) secretory IgA promotes eosinophil survival through autocrine GM-CSF [88]. Overlapping signaling outcomes, abundant tissue expression and extended eosinophil responsiveness to GM-CSF over IL-5 may influence the effectiveness of IL-5-targeting biologics to deplete intestinal tissue-resident, activated eosinophils [89]. Within the context of inflammation, localized tissue factors including tissue alarmins, immune cell-derived inflammatory cytokines, antigen-bound immunoglobulins, and local hypoxia further shape eosinophil phenotypes, metabolic profile and function, temporally and spatially, within intestinal tissues [69,88] (Figure 2).

#### 4.2.2. Extracellular Matrix

“Inside-out” signaling refers to the regulation of expression and functional conformation of surface integrins. As described in the above sections, the local tissue environment shapes eosinophil integrin and adhesion molecule expression, including levels and active conformations of heterodimers involving CD11c and CD11b. Modulations of eosinophil adhesion molecules thereafter further shape eosinophil activation and spatial functioning in situ through “outside-in” signaling generated through engagements between eosinophil-expressed integrins and extracellular matrix (ECM) components.

The ECM is a three-dimensional network of bioactive polymers, composed of triple helical collagens and complex proteoglycans whose composition and intermolecular interactions are in constant flux in response to the tissue microenvironment. Therefore, the ECM provides not only structural support and scaffolding, but also serves as a highly dynamic signaling network, integrating mechanical, electrical and chemical environmental cues, and communicating these signals to immune cells (reviewed in [90]). ECM interactions influence immune cell adhesion, proliferation, migration, survival and function [90,91]. Eosinophils can engage directly with ECM matricellular and structural components, including (but not limited to) tenascins, hyaluranon, periostin, fibronectin, laminin and collagens, through interactions with surface-expressed adhesion molecules including CD11b and CD11c [58,92] (Figure 2).

Of particular interest in eosinophilic inflammatory diseases is the 90kDa ECM-associated matricellular protein periostin. First recognized for its contributions to cardiac tissue repair and remodeling [93,94], periostin is also induced by mechanical stress [95] and mediators associated with type 2 immunity, in particular IL-13 [63]. Periostin is linked to eosinophil recruitment to inflamed airways in asthmatics and idiopathic pulmonary fibrosis (IPF) patients; serum periostin predicts airway eosinophils in steroid-resistant asthma [96], and predicts disease progression in IPF [97]. Periostin is likewise highly upregulated in the esophagus of EoE patients [38,63], and in a mouse model of EoE, periostin deletion was associated with defective eosinophil esophageal recruitment [63]. Beyond regulating eosinophil recruitment, periostin supports α_M_β_2_ (CD11b/CD18)-mediated adhesion and migration of IL-5, GM-CSF or IL-3 stimulated eosinophils, promotes eosinophil adhesion to fibronectin, prolongs eosinophil survival, and enhances eosinophil superoxide anion production and secretion of EDN [38,62,63,87,98]. Productive interactions driven through eosinophil-expressed α_M_β_2_ are particularly relevant, as tissue eosinophils recruited to the airways of asthmatics and the esophagus of EoE patients exhibit a higher percentage of α_M_β_2_ in activated conformations [38]. Periostin-driven effects on eosinophil adhesion and migration were highly dynamic and dose-dependent, with optimal conditions (i.e., intermediate concentrations of periostin) eliciting a polarized, “acorn-shaped” eosinophil morphology with intracellular granules organized together within the cytoplasm and rapid, consistent migration. In contrast, suboptimal concentrations of periostin (including low or very high concentrations) elicited a flattened, “pancake-like” morphology with dispersed intracellular granules and slower chemokinetic movements [87]. This same study also revealed a hierarchy in priming potencies of eosinophil-activating cytokines, with GM-CSF eliciting eosinophil adhesion to periostin at lower concentrations than those required for IL-5 or IL-3 [87]. Similar to most other immune cells, communication between eosinophils and the ECM are bi-directional. Eosinophil-derived cytokines, peroxidase and metalloproteinases influence fibroblast function and ECM composition, and regulate ECM turnover and biosynthesis [99,100,101,102].

#### 4.2.3. Receptors Regulating Survival and Activation of Intestinal Eosinophils

Eosinophils within intestinal tissues exhibit much longer life spans (up to at least 14 days) compared to lung eosinophils (<36 h) [55]. Supported by the intestinal tissue milieu, multiple factors contribute to the enhanced longevity of intestinal eosinophils, including GM-CSF-mediated signaling [103], autocrine survival signals [104], interactions with extracellular matrix components [38,105], and signaling through the highly-expressed γ_c_-chain [55].

Counteracting the pro-survival and activation signals inherent in the cytokine milieu, intestinal eosinophils express elevated levels of several inhibitory receptors, including Siglec 8 (human)/F (mouse), SIRP1α, CD22, CD300a/f, and PIRB. Siglec (sialic acid-binding immunoglobulin-like lectin)-8/F is an inhibitory single-pass transmembrane receptor that interacts with a sulfated glycan ligand to regulate eosinophil survival [106]. SIRP1α (inhibitory receptor signal regulatory protein α/CD172a) is an ITIM-containing receptor highly expressed on intestinal eosinophils that inhibits activation-induced degranulation of eosinophils, likely through binding its ubiquitously-expressed ligand, CD47 [107]. CD22 is an inhibitory pan-B cell marker belonging to the family of Siglec (sialic acid-binding immunoglobulin-type lectin) receptors that negatively regulates the strength of B cell receptor signaling. CD22 is not detected on eosinophils from blood, spleen or thymus, but is induced on lung and intestinal eosinophils, with 10-fold higher expression on the latter. Genetic ablation of CD22 resulted in higher numbers of intestinal eosinophils, suggesting CD22 negatively regulates intestinal eosinophilia [64]. CD300a and CD300f are expressed by human and mouse eosinophils and can bind phospholipids associated with cell death, such as phosphatidylserine, phosphatidylethanolamine and ceramide (reviewed in [108]). CD300a dampens both GM-CSF- and IL-5-elicited survival signals and eotaxin-driven chemotaxis [109], and is further upregulated by hypoxia [110]. CD300f is also a negative regulator of CCR3 signaling, preventing eosinophil chemotaxis. Importantly, in addition to immunoreceptor tyrosine-based inhibition motifs (ITIMs), CD300f expresses additional intracellular docking motifs that allow it to also function as a co-activator. Co-activating effects of eosinophil-expressed CD300f include enhancing signaling downstream of PRR engagement, ST2 receptor complex activation and IL-4-induced STAT6 signaling, exacerbating chemokine secretion [108]. Like CD300a/f, PIRB (paired immunoglobulin-like receptor B) is a negative regulator of eotaxin-induced chemotaxis, although in contrast, PIRB enhanced LTB_4_-mediated eosinophil responses [111]. PIRB was upregulated by IL-13 on esophageal eosinophils in a mouse model of EoE, and inhibited IL-13-mediated activation. Transcriptional analyses of PIRB-deficient esophageal eosinophils revealed diminished expression of pro-fibrotic genes, suggesting PIRB functions as a checkpoint inhibitor of eosinophil adoption of a pro-fibrotic phenotype [112].

Together these data reveal a unique surface receptor phenotype of intestinal eosinophils, shaped by local environmental signals, that drives and restrains eosinophil activation, survival and function. These receptors and/or their tissue and cellular ligands may identify novel targets for therapeutic modulation of eosinophil activation, survival and function. To this end, a humanized afucosylated antibody against Siglec 8, demonstrated to induce eosinophil death and inhibit mast cell degranulation, is in clinical development for eosinophil-mediated diseases [113,114,115].

#### 4.2.4. Systemic Immune, Metabolic and Hormonal Signals Contribute to Intestinal Eosinophil Regulation

The progression of Th2 allergic conditions (e.g., atopic dermatitis, asthma and food allergies) associated with the “allergic march” is likely driven in part by systemic Th2-associated mediators [116]. Within the context of the allergic march, we recently demonstrated that the frequency and phenotype of resident intestinal eosinophils are regulated by as yet undefined signals generated through mucosal organ crosstalk along the lung:gut and skin:gut axes [67].

As noted in Section 2 above, adipose tissue eosinophils are implicated in regulation of glucose metabolism and metabolic homeostasis in the steady state [117,118]. Conversely, eosinophils are also regulated by systemic metabolic and hormonal signals. Circadian variation in the concentration of blood eosinophils was first noted in the 1930s and is aligned with levels of serum IL-5 and regulated by light/dark cycles, the adrenal–cortical axis and, most predominantly, by the timing of caloric intake [119,120,121]. Nussbaum etal. provide evidence for a mechanistic basis for nutrient-dependent regulation of blood and intestinal eosinophil numbers under homeostatic conditions; i.e., feeding elicits secretion of the peptide hormone vasoactive intestinal peptide (VIP), which in turn signals through VPAC2 receptors on long-lived intestinal ILC2 cells, promoting secretion of IL-5 [77]. Psychological stress has likewise been linked to alterations in peripheral eosinophil counts [122] and modulation of intestinal eosinophil function [123], as stress-induced substance P elicits secretion of corticotrophin releasing hormone (CRH) from jejunal eosinophils, promoting intestinal barrier dysregulation indirectly through mast cell activation [123]. Eosinophils express multiple additional hormone and metabolite receptors, including receptors for leptin [124], estrogen [125], retinoic acid [126,127], prostaglandins and leukotrienes [128,129], purine nucleotides [130], and fatty acids [131,132,133], further supporting their potential roles as endocrine, nutrient and metabolic sensors in health and disease.

#### 4.2.5. Impact of the Microbiome on Intestinal Eosinophils

Genetic factors, diet, clinical intervention (e.g., antibiotics), and microbial pathogens modulate the load and diversity of the microbiome in health and disease. Dysregulation of the microbiome in turn profoundly influences both innate and adaptive arms of the mammalian immune system. Direct interactions between microbes and toll-like receptor (TLR)- or other pattern recognition receptor (PRR)-expressing immune cells, including eosinophils (reviewed in [134]), can occur in situations of epithelial barrier breach. Direct interactions with bacteria evoke multiple functional responses from eosinophils (reviewed in [135]) including phagocytosis, release of granule-derived proteins, reactive oxygen species (ROS) generation, and extrusion of extracellular “traps” into the tissue, the latter consisting of DNA strands and granule-derived cationic proteins [7]. With an intact barrier, microbial regulation of immunity is still accomplished, in large part by signaling through immune-cell-expressed G-protein-coupled metabolite sensors including free fatty acid receptors (FFARs) [132,133,136]. Intestinal eosinophils and their upstream cellular regulators (e.g., ILC2s) express FFARs and respond to microbial metabolites [132,137]. Elimination of endogenous microflora, either naturally in germ-free mice or artificially through antibiotic treatments, elicits higher total numbers of intestinal eosinophils with concomitant alterations in eosinophil granularity and morphology associated with the modulation of eosinophil attraction and retention signals, a situation reversed with reconstitution of the microbial flora [138]. Less extreme alterations in the composition of the microbiome and/or diet-derived fuel sources also fine-tune eosinophil tissue levels. For example, increased abundance of commensal microbes such as *Ruminococcus gnavus* promotes eosinophil infiltration of the lung and colon of infants [139]. In contrast, inverse correlations between levels of tissue eosinophils and short chain fatty acids (SCFAs, e.g., butyrate, propionate), metabolic products of microbial fermentation of indigestible dietary fibers, have been described in allergic diseases of the airways [136,140,141] and GI tract [136], with SCFAs acting through direct [136] and indirect (e.g., via ILC2s or modulating pH) [136,137,140] mechanisms to modulate eosinophil recruitment, function and survival. Of note, modulation of the intestinal microbiota appears to impact not only intestinal eosinophils, but also eosinophils resident in distant tissues. For example, loss of the endogenous microflora promoted eosinophil recruitment into inguinal subcutaneous tissue in association with beige fat development [142], likely driven by eosinophil-mediated macrophage polarization [118,143]. Growing evidence suggests important roles for the microbiome of the upper gastrointestinal tract as well. Several studies have demonstrated dysregulation of the salivary [144] and esophageal [145,146,147,148,149] microbiomes in patients with eosinophilic esophagitis (EoE), including changes in bacterial load and composition, correlating with disease activity. How the esophageal or intestinal microbiomes might impact disease susceptibility, progression or amelioration in EoE or other EGIDs remains to be determined.

Cross-influences between the microbiome and immune cells, including intestinal eosinophils, are bi-directional. Intestinal eosinophils support the protective mucosal barrier in steady state; in some studies (but not all [21]), absence of eosinophils is associated with baseline alterations in the intestinal mucus layer and the maintenance of IgA-expressing plasma cells, particularly in the small intestine [19,20,22]. Although the resultant microbial profiles differ between studies, genetic ablation of eosinophils elicited substantial shifts in microbial diversity compared to wild-type mice, especially among mucus-resident bacteria [19,20,21]. Following barrier breach, intestinal eosinophils engage in direct bactericidal activities through secreting anti-bacterial factors [18] and expelling extracellular DNA traps that sequester and destroy bacteria [7], and in *Clostridium difficile* infection, eosinophils are implicated in IL-25-driven barrier protection [150,151]. Therefore, intestinal eosinophils impact barrier integrity, directly and indirectly participate in anti-bacterial immunity, and influence the composition and load of the commensal microbiota.

Together, these data are revealing complex bi-directional relationships between the microbiome and tissue eosinophils that may have profound implications on disease susceptibility. Further studies are needed to unravel eosinophil–microbiome cross talk, and to consider microbiome modification (e.g., dietary, pre- or pro-biotic approaches) as a strategy to shape tissue eosinophil phenotypes.

## 5. Key Unanswered Questions and Potential Implications for Therapeutic Approaches

Several biologics currently in use or in the developmental pipeline directly or indirectly impact eosinophil numbers and/or functions and have been carefully reviewed elsewhere [152,153]. These include targets associated with Th2 immunity (e.g., IL-4, IL-13, IgE, and TSLP) and targets associated more specifically with eosinophils (e.g., eotaxin, CCR3, PGD2, Siglec 8, and IL-5/IL-5Rα). As mentioned in Section 4.2.3 above, a non-fucosylated antibody targeting Siglec 8 (lirentelimab) has been shown to induce eosinophil cell death and inhibit mast cell degranulation. An in-depth review of Siglec 8 and its promise as a therapeutic target is included elsewhere in this journal issue [114]. Phase 2/3 studies of lirentelimab are currently underway in EGIDs (ENIGMA 2; NCT04322604 and KRYPTOS; NCT04322708). In line with the thematic focus of this journal issue (i.e., “Eosinophils beyond IL-5”), we limit our discussion here specifically to biologics targeting the IL-5 pathway in EGIDs.

### 5.1. Anti-IL-5 Biologics in EGIDs: Successes and Shortcomings

IL-5 is a critical eosinophilopoietin, driving the expansion of eosinophil-committed progenitors within the bone marrow and priming mature eosinophils for enhanced function and survival [154]. Anti-IL-5 therapeutics are now approved for severe eosinophilic asthma and have been studied in several eosinophilic-mediated diseases. Targeting IL-5 in the treatment of EGIDs has been the subject of several clinical trials, specifically in EoE, wherein chronic Th2-driven inflammation leads to esophageal remodeling and symptoms of esophageal dysfunction, classically intermittent solid food dysphagia. In clinical trials in both adults and children with EoE, anti-IL-5 biologics have been able to reduce mucosal and peripheral eosinophilia but have often missed their primary aims with regards to degree of eosinophil reduction or symptom improvement.

Mepolizumab is a humanized anti-IL-5 monoclonal IgG1 antibody that binds to IL-5, preventing binding to IL-5Rα on the surface of eosinophils [155]. It has been studied in adults and children with EoE [156,157,158]. In each of these studies, mucosal eosinophilia was substantially decreased compared to placebo, but their ability to reach their primary end points was limited. While there was a reduction in mean mucosal eosinophilia (54% compared to 5% in the placebo group), no patients reached the primary end point of <5 eos/hpf and no significant change in patient-reported symptoms was detected [156]. Symptom detection was performed using a published but non-validated score focusing on esophageal symptoms in EoE [159]. The ability to adequately assess esophageal outcomes and symptoms in EoE has challenged several clinical trials in EoE [156,157,158,160]. In a randomized double-blind parallel-group trial in children and adolescents mean mucosal eosinophilia was again reduced significantly in the study group, but only a small portion (8.8%) reached the histologic primary outcome of a peak eosinophil count of <5 eos/hpf [158]. No significant difference in symptoms was detected; however, it was noted that patients had low values of symptom scores at baseline.

Reslizumab is also a humanized anti-IL-5 monoclonal antibody that prevents binding to IL-5Rα. In a randomized double-blind placebo-controlled trial in over 200 children and adolescents, reslizumab reduced peak esophageal eosinophilia compared to placebo. All participants demonstrated an improvement in EoE symptom severity with no difference between participants that received reslizumab or placebo [160]. Again, mild symptoms in some patients at study entry and lack of validated symptom measures may have affected the investigators’ ability to capture symptom improvement. Alternatively, the degree to which anti-IL-5 treatment reduces mucosal eosinophilia alone may not be sufficient to improve symptoms. Benralizumab is a humanized monoclonal antibody against the IL-5Rα subunit [161]. The drug induces NK cell-mediated killing of target cells through an antibody-dependent cell-mediated cytotoxicity [162]. Benralizumab, approved for use in severe eosinophilic asthma, is also currently under investigation in EoE and Eosinophilic Gastritis (NCT04543409; NCT03473977).

Studies to date that have evaluated the efficacy of therapeutics targeting IL-5 were performed rather early in the evolution of standardizing clinically meaningful measurements of esophageal and patient-reported outcomes. In addition, inclusion criteria varied with regards to symptom severity at baseline, allowed use of other medications and history of failed treatments. Given positive findings with regards to improved mucosal eosinophilia and other molecular targets, we may find it worthwhile to revisit this therapeutic class in the treatment of eosinophilic GI diseases as measures of clinical outcomes improve. Greater insight into endotypes within the spectrum of eosinophilic GI diseases will likely help us target patient groups that are most likely to benefit from specific approaches to treatment, whether they be with anti-IL-5 or other biologic classes targeting eosinophilic inflammation.

### 5.2. Key Knowledge Gaps and Potential Implications on Eosinophil-Targeting Strategies

Growing evidence in health and disease points to heterogeneity in eosinophil functional sub-phenotypes, shaped by systemic and local tissue signals. To the extent that beneficial versus detrimental functions of eosinophils might be attributable to these distinct sub-phenotypes, defining their distinguishing proteomic, transcriptomic, surface receptor and metabolomic profiles therefore remains an important objective. Filling these knowledge gaps may provide important insights into understanding the paradoxical contributions of eosinophils in health and disease and inform next-generation pharmacotherapeutic approaches designed to restrain inflammatory eosinophil functions while sustaining eosinophil homeostatic roles.

## Figures and Tables

**Figure 1 cells-10-00426-f001:**
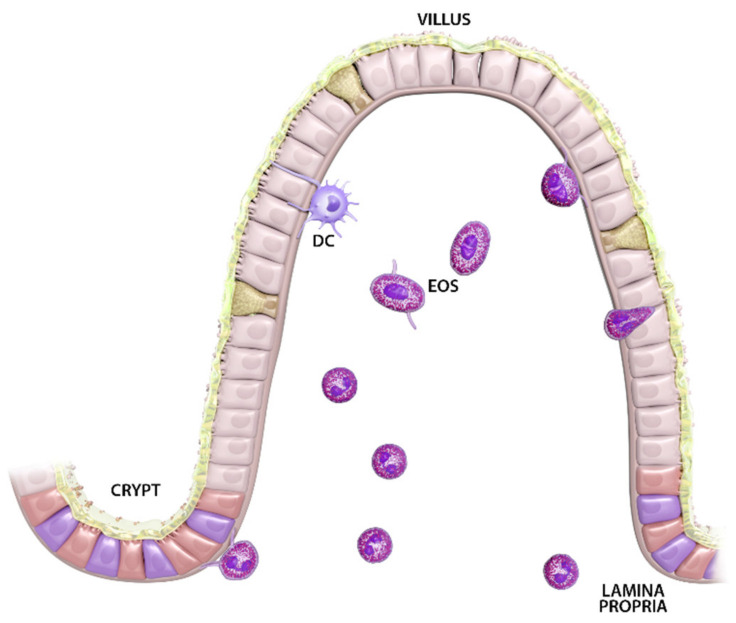
Lamina propria (LP) eosinophils display compartmentalized morphological heterogeneity in the steady state. Eosinophils localized within the LP surrounding crypts appear round, with few, if any, short cellular extensions. In contrast, eosinophils localized to LP within villi often display spheric nuclei, elongated cell bodies and more commonly exhibit cellular protrusions. Eosinophils are frequently observed lying along the villus epithelium, extending cellular processes that breach the basement membrane. A small subset of eosinophils, expressing elevated levels of surface CD11c, CD11b and Siglec F, is also detected within the epithelial layer.

**Figure 2 cells-10-00426-f002:**
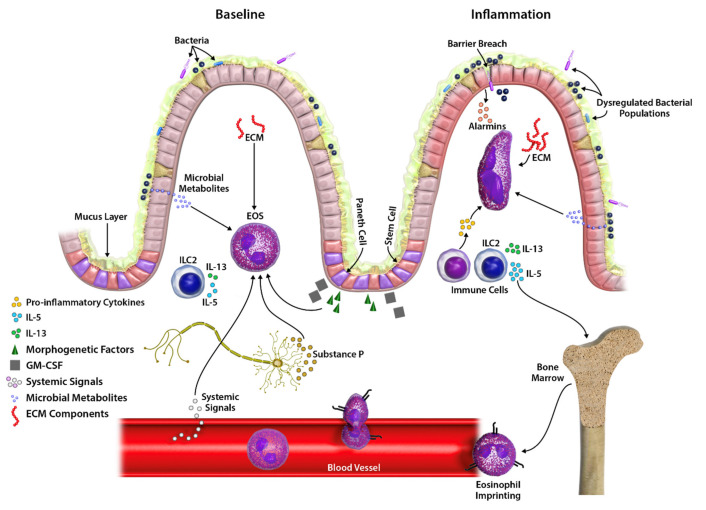
Pressures that influence intestinal eosinophil function and phenotype in health and disease. Eosinophil recruitment, accumulation and functional phenotypes in steady state and inflammation are shaped by complex factors, including systemic hormonal and metabolic mediators, the microbiome, the local cytokine milieu, interactions with extracellular matrix components, tissue factors, and epithelial- and immune-cell-derived cytokines. EOS, eosinophil.

## Data Availability

No new data were created or analyzed in this study. Data sharing is not applicable to this article.

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
