# Peer review of "Heterogeneity of Intestinal Tissue Eosinophils: Potential Considerations for Next-Generation Eosinophil-Targeting Strategies"

_cells, 2021, doi:10.3390/cells10020426_

Round 1
Reviewer 1 Report
- Consider closely the use of the word “Plasticity”. This is generally kept to cells that change transcription factor expression that defines their phenotype that can be induced to change from one to the other. For example T cells. Rather the literature shows sub-phenotypes can be activated by specific stimuli. No one has shown that sub-phenotype A can now become sub-phenotype B definitively to date. For example, a regulatory eosinophil in the lung- has not been shown to become inflammatory or visa versa. I am not discounting that studies will show this clearly someday, I am just not sure it is clear at this time.
- Subgroups, sub-phenotype, etc. Maybe provide a sentence to explain what does subtype or sub-phenotype or subgroup mean with regards to this review.
- Change EPO to EPX
- Sentences 70-75. Break down that long sentence into 2 or 3 for flow. Writing style note: The authors have a tendency on occasion to write sentences with about 4-7 serial points or examples separated by commas or semicolons and it can get a bit confusing-I will highlight a few below.
- By line 78 I wanted the authors to just jump into the main topic without what felt like a slightly redundant intro sentence to above it.
“Although eosinophils hold well characterized terminal effector functions involving 78 anti-helminthic and pro-allergic functions, eosinophils play an underappreciated and 79 understudied role in both tissue homeostasis and anti-inflammatory processes.” Maybe just go: “Eosinophils play an underappreciated and 79 understudied role in both tissue homeostasis and anti-inflammatory processes.”
- “These observations prompted Lee and col-88 leagues to propose an alternate conceptual framework, termed the LIAR (Local Immun-89 ity And/or Remodeling/Repair) hypothesis, that posits eosinophils are primarily tissue 90 repair cells, attracted to areas of focal surges of cell death and proliferation in an attempt 91 to restore tissue homeostasis [14].”
The theme of LIAR per my interpretation of the literature: “Posits eosinophils are primarily immune cells that also have reparative properties….”
- Line 96. Readers may not know what DNA “nets” means.. .. Assuming referring to DNA extracellular traps that form a netlike structure to kill bacteria. This is indeed described later in the review.
- Line 98 “mucous secretion” Do those papers show reduction in mucins and related release from epithelial cells? I thought they were mostly IgA and microbiome? This is a key distinction for those studying direct effect of immune cells on mucins (Muc5, eg.) in mucosal tissues.
- Technical to journal: Will you use the Greek form for things like IL-1b should be IL-1β
- Another long sentence. It reads as if protectin D1 contributes to 12/15-lipo. Separate into two sentences to distinguish that the protectin D1 from 12/15 lipoxygenase. This review is allergy, but pretty cool if it helps https://doi.org/10.1016/j.prostaglandins.2020.106477
- Line 155. I do not agree with Siglec-Flo or am concerned on interpretation by non-eosinophil readers. Siglec-F is still fairly high on eosinophils in the blood relative to any other cell. This may give the wrong impression of nearly being negative in expression. Siglec-Fint would be more appropriate maybe. Siglec-8 is also increased MFI relative to controls/other cells in the blood.
- 159 “temporarily imprint” imprint to me means epigenetics, which may not necessarily be temporary. Can you explain what you mean here in a different terminology?
- Are there examples of blood eosinophil cell surface expression changing with EGIDs? An example to give here in addition to the asthma examples?
- Paragraph along 170: The pre-formed granules is only known for human eosinophils. A note should be made we do not know the extent of pre-formed granules in mouse eosinophils. Also, this may give the impression eosinophils do not actively transcribe in response to stimuli- yet both mouse and human eos do transcribe new cytokine mRNA.
- Line 205: Check out this reference PMID: 19843944. Your reference 53. It is one of the earlier (2009) to show cell surface and HE shape per tissue (siglec-F shown for lung, thymus, intestine, etc.).
- Consider: PMID: 28095479. Used Aspergillus like Rosenberg’s. Different read out.
- Line 257, Remove sentences 257-260 as sentence 264 is sufficient to get this point across and the other two are out of sync as they jump into lung too much.
- Line 268; I would like to see a mention of the cytokine receptors that use gc. (IL-2, IL-4, IL-7, IL-9, IL-15 and interleukin-21 receptor) as I think this is underappreciated by many. Do we know the expression of PIRB in the intestine? It is mentioned later, but not here.
- Line 308 area: you highlight regional differences of eos in the intestine. Does any of this associate with natural microbiome? Bacteria composition changes from small to large, nutrients (SCFAs) change from small to large, etc. Is it worth a few sentences before going into the disease settings next?
- OK I see it in section 4.2.5. Consider putting this right after the introduction of homeostatic eosinophils before section 2.3 as this is a homeostatic function and highlights their diverse roles in the intestine, yet does not have a lot of “heterogeneity” detail as much as the disease and features of eos later.
- Line 320 Note PMID: 31856334 in addition to REF 64.
- Line 368, another long sentence where it isn’t entirely clear you mean eotaxin-1 acting through CCR3. Can you re-write.
- Figure 1. Is there a way to zoom in a bit more and show the shape change better for the subtypes of eosinophils- get a better DC like arm for one of them through the epithelial layer. Maybe highlight the cell surface expression on the cells per situation?
- Figure 2: Again, zoom in more to the intestinal wall and make the blood vessel and bone smaller. For dysregulated bacterial populations- can you make them aa different color than the ones on the left to just visually separate the two?
- Line 470. Eosinophils in the lung may be 36 hrs at homeostasis but not inflammation. Maybe clarify again this is homeostatic timing. Section 4.2.3 is nice for this reviewer-good description of the cell surface receptor functions and details.
- Conclusions area on biologics, Add a mention of Silgec-8 antibody. I believe this is also being tested in an EGID?
- Conclusion sentence. I go back to the use of the word “plasticity”. Consider closely if this is what you mean to truly say here.
Very nice review.
Reviewer 2 Report
Congratulations on the well written review.
The article is a narrative review, it is very well written, it is broad and optimally covers all the fields in which eosinophils are involved, it has both a basic research slant and an applicative slant (it also mentions potential drugs like anti-interleukin 5).
I really couldn't have thought of how to write it better and in my opinion it has no weaknesses.
Obviously ,I repeat, it is a narrative review.
Author Response
Thank you for reviewing this manuscript, and for the positive comments.